# SoundAct: Learning Spatial Sound Awareness for Egocentric Robot Manipulation with Stereo Audio

*Abstract*— **Humans naturally use auditory and visual cues to interact with objects beyond sight. However, most robot manipulation frameworks rely solely on vision, limiting their ability to handle audio-driven tasks (*e.g.*, alarm clock turn-off) and out-of-view events. To address this, we propose `SoundAct`, a spatial sound-aware egocentric robot manipulation framework that integrates stereo microphones with an egocentric camera for beyond-sight spatial audio reasoning. We encode directional cues from stereo audio as magnitude spectrograms and fuse them with visual features via an attention mechanism, enabling the policy to jointly reason over auditory and visual cues. We further introduce a spatial audio augmentation method to improve robustness under audio distractors. We evaluate our method on a *beyond-sight ring-off* task and demonstrate effective manipulation of sound-source objects beyond sight. Video is available on https://drive.google.com/file/d/1i-LP_FzB9-oS55BpKOD9HS2qL9adld81/view?usp=sharing.**

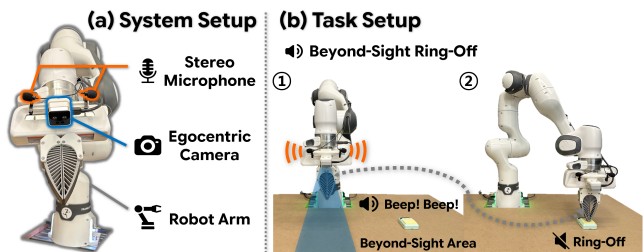

Fig. 1: **Overview of `SoundAct`.** (a) System setup with a stereo microphone setup and a wrist-mounted egocentric camera on a robot arm. (b) Task setup for spatial sound-aware manipulation, where the robot localizes a ringing alarm clock outside its field of view using stereo audio and turns it off once it becomes visible.

## I. INTRODUCTION

Humans jointly use spatial auditory awareness to interact with objects out of sight, such as turning off the ringing alarm clock hidden from the field of view (FOV). In contrast, many robot manipulation policies rely heavily on visual observations, often failing when events occur outside the FOV of the camera. Recent multimodal approaches have incorporated audio but typically treat it as a tactile signal [1]–[3] or as contextual information [4], rather than as a spatial cue for guiding action. Although sound source localization has been studied in robotics [5], [6], its integration into robot manipulation policy learning remains limited. Consequently, sound-based spatial awareness in robot manipulation is still underexplored.

To address this, we propose SoundAct, a spatial sound-aware egocentric robot manipulation framework that integrates stereo microphones with a wrist-mounted egocentric camera on a robotic arm. As illustrated in Fig. 1-(a), our system leverages stereo auditory cues to implicitly capture the direction of the sound source occurring beyond the FOV of the camera and incorporates them into policy learning, enabling the robot to respond to beyond-sight auditory events.

In this work, two key questions arise in enabling spatial sound-aware manipulation. First, how can stereo audio be represented to capture spatial cues for action reasoning? We address this by encoding stereo audio as the magnitude of spectrograms using the short-time Fourier transform (STFT), enabling the policy to implicitly infer the direction of sound sources beyond the visual field. Second, how can the policy robustly focus on the target audio under distractors? We introduce a spatial sound augmentation method that independently scales noise audio applied to the left and right

channels. This channel-specific scaling enables the policy to remain focused on the target audio, even in the presence of louder audio distractors.

To evaluate the spatial sound awareness of our framework, we introduce a *beyond-sight ring-off* task, in which the robot first moves toward a ringing alarm clock outside its visual FOV using stereo audio and then turns it off once the object becomes visible, as illustrated in Fig. 1-(b). Our results show strong in-distribution performance, while also demonstrating that the policy can distinguish target audio from distracting sounds, jointly leverage audio and vision in cluttered scenes with decoy objects, and remain robust to unseen initial poses thanks to the broad spatial coverage provided by audio.

## II. RELATED WORKS

### A. Audio-Guided Robot Manipulation

Visual policy learning methods such as Diffusion Policy [7] predominantly rely on fully observed third-person views, and even egocentric approaches suffer from severe occlusion and limited FOV. To overcome these visual limitations, recent works incorporate audio into manipulation policies, either as a contact signal [1]–[3] or as contextual information [4], [8]. However, both directions do not exploit audio as a directional spatial cue. To our knowledge, SoundAct is among the first real-world egocentric manipulation frameworks to leverage stereo audio as a spatial cue for beyond-sight action.

### B. Spatial Audio Awareness in Robotics

Microphone arrays are often used for sound source localization [9]–[11], and in robotics, this has been primarily applied to audio-visual navigation. SoundSpaces [6] and SAVi [12] leverage binaural audio egocentrically to compensate for visual limitations, but remain confined to simulated

navigation with reinforcement learning, where only coarse directional movement is required. *SoundAct* brings stereo spatial audio awareness to real-world robot manipulation with behavior cloning, where precise actions are essential.

## III. METHOD

### A. Egocentric System Setup

**Hardware setup.** Our hardware platform includes a Franka Research 3 robot arm and gripper integrated with a wrist-mounted Intel RealSense D405 camera. For auditory perception, two omnidirectional Maono AU-XLR10 condenser microphones are mounted laterally relative to the viewing direction of the camera, capturing stereo audio via a Behringer UMC404HD audio interface.

**Data collection setup.** Raw sensory streams, including wrist-view video ($60\,\mathrm{Hz}$, $640{\times}480$), stereo audio ($48\,\mathrm{kHz}$), and robot states (end-effector pose $100\,\mathrm{Hz}$ and gripper state at $30\,\mathrm{Hz}$), are synchronized and resampled to a unified rate of $20\,\mathrm{Hz}$.

### B. Policy Architecture

As depicted in Fig. 2, we design a multimodal policy that integrates visual, auditory, and proprioceptive observations to enable manipulation both within and beyond sight. The motivation is that vision alone is unreliable under occlusion or limited FOV, while stereo audio provides complementary cues about a sound source beyond sight.

**Data preprocessing.** During training, the policy uses a 2-step observation of images resized to $224{\times}224$ and end-effector poses, together with 2-second stereo audio resampled to $16\,\mathrm{kHz}$. This short temporal window captures transient auditory events while keeping the input dimensionality tractable.

**Vision encoding.** Egocentric images are processed using a ViT encoder [13] initialized with CLIP pretrained weights, which provides strong semantic priors that generalize well across diverse object appearances and scene configurations. Following [1], we extract the [CLS] token from each frame in the observation window and concatenate them along the temporal dimension to form the visual latent features. This design retains a compact yet expressive representation of the scene while preserving short-term temporal dynamics that are useful for reactive control.

**Stereo audio encoding.** For robust manipulation, the audio representation should preserve spatial directional cues for localizing the target sound, while remaining invariant to the various ambiguities of spatial audio, such as reflections, reverberation, and phase distortions that arise from environmental and geometric variations. To this end, we transform the raw stereo waveform into time–frequency representations via the STFT, retain only the magnitude spectrograms of both channels, and stack them as a 2-channel input to a modified ResNet-18 encoder. The encoded features are then projected through a lightweight MLP to match the dimensionality of the visual latent features. This design preserves inter-channel level differences that provide directional cues for source localization, while avoiding the unstable phase components

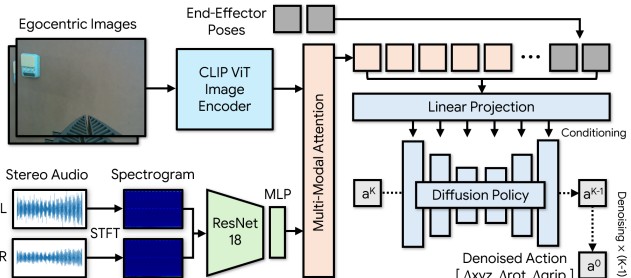

Fig. 2: **SoundAct** model architecture.

introduced by 4-channel magnitude–phase formulations [14] and the loss of fine-grained spatial cues caused by log-mel representations [1].

**Feature fusion.** The visual and audio latent features are concatenated along the token dimension and processed by a Transformer encoder with self-attention [15]. The self-attention mechanism allows the policy to dynamically weight visual and auditory cues depending on context, for instance, attending more strongly to audio when the target is occluded and more strongly to vision when the target is clearly visible. The fused multimodal representation is then concatenated with the end-effector pose to incorporate proprioceptive state, and serves as the conditioning input for the downstream action prediction module.

**Action prediction.** Robot actions are predicted using the Diffusion Policy framework [7], in which a 1D convolutional UNet iteratively denoises action sequences conditioned on the fused multimodal representation. Starting from Gaussian noise, the UNet performs k denoising steps to produce action trajectories, enabling the policy to model multimodal action distributions and capture temporally coherent behaviors. The policy is trained with a mean squared error (MSE) loss, corresponding to supervision on relative action trajectories.

**Spatial audio augmentation.** A key challenge in stereo audio-conditioned manipulation is that the policy can over-fit to spurious cues such as absolute loudness, rather than the inter-channel cues that indicate source direction. While ManiWAV [1] injects background noise for contact-microphone settings, naively applying this to stereo audio leaves directional cues intact, as identical noise on both channels does not perturb them. To address this, we extend background noise injection to the spatial setting: with probability $p$, we sample a distractor noise clip $n$ from ESC-50 [16] and mix it into the stereo stream with complementary per-channel scaling as

$$\tilde{x}_L = x_L + s{\cdot}n, \quad \tilde{x}_R = x_R + (1-s){\cdot}n, \quad s \sim \mathcal{U}(0,1), \quad (1)$$

where $x_L, x_R$ denote the left and right channels of the original stereo audio. The complementary scaling preserves the overall energy of the distractor while varying its inter-channel balance, effectively simulating distractors with varying left-right dominance relative to the microphones. As a result, the policy is exposed to diverse spatial configurations of distractors during training, improving its ability to disen-

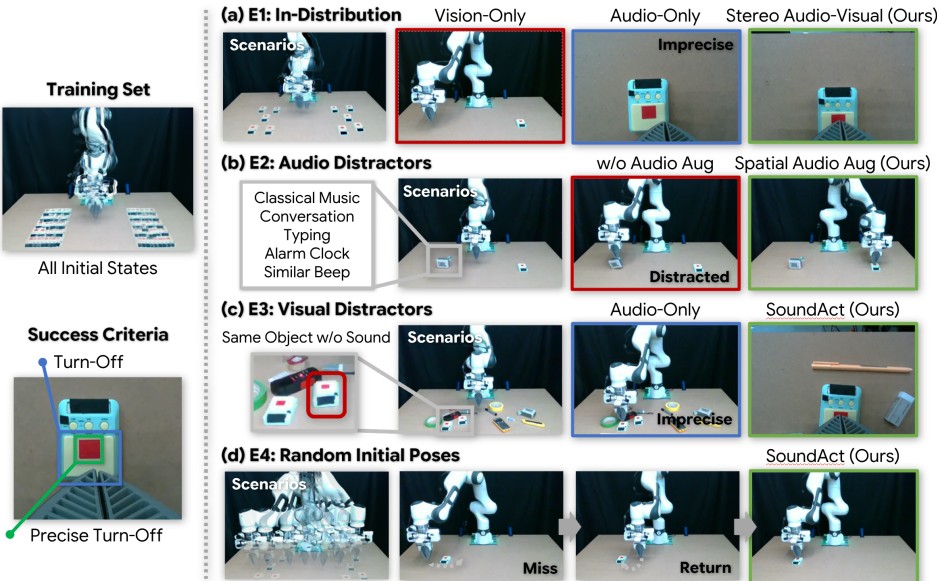

Fig. 3: **Beyond-sight ring-off evaluation.** Trained in a distractor-free setting, the policy is evaluated in four scenarios to assess its robustness. A third-person camera is used only for visualization.

tangle the target source from competing sound sources and to remain robust to variations in absolute sound magnitude at deployment.

## IV. EXPERIMENTS

### A. Implementation Details

We convert stereo audio into magnitude spectrograms using STFT (FFT=512, window=400, hop=160). Visual and audio features are encoded into $768 \times 2$ dimensions and fused into a 768-dimensional representation. We apply random crop (0.95) and color jitter for visual augmentation, and audio augmentation with a probability of 0.5. The policy network is optimized using AdamW with a batch size of 64 over 50 epochs on a single RTX PRO A6000 GPU, with EMA. For the diffusion process, we adopt DDIM [17] scheduling with 50 denoising steps during training.

### B. Task Setup

**Data Collection.** We evaluate our framework on a *beyond-sight ring-off* task, where the robot infers the direction of an alarm clock outside its initial FOV and navigates toward it using stereo audio, turning it off once visible. For training, we collect 80 teleoperated episodes with the target uniformly placed within a $40\,\mathrm{cm} \times 20\,\mathrm{cm}$ rectangular region on either the left or right beyond-sight area.

**Evaluation.** Fig. 3 and Table I present the qualitative and quantitative results, respectively. We evaluate each setting over 10 trials and report the mean and standard deviation across three models trained with different random seeds. In all evaluations, Success Rate (SR) measures whether the robot successfully turns off the target alarm clock within the time limit of approximately 14 seconds, while Precise Success Rate (PSR) counts only cases in which the robot accurately presses the red tag within the same time limit, as illustrated in Fig. 3.

### C. In-Distribution Evaluation

Fig. 3-(a) and Table I-(a),(b) report the in-distribution results. We evaluate 10 trials with alarm clocks placed in the similar spatial distribution as in training, with five targets on the left and five on the right.

**Modality comparison.** As shown in Table I-(a), Vision-Only performs poorly because the target initially lies outside the camera FOV, providing no information for directional search. Audio-Only with stereo microphones achieves a high SR, indicating that stereo audio alone is sufficient for target localization. However, its PSR remains low, showing that audio alone is insufficient for precise turn-off near the target. Audio-Visual (Mono) improves PSR by using vision for the final interaction, but its overall performance is limited by the weak directional cue from a single microphone. In contrast, our Stereo Audio-Visual policy achieves the best SR and PSR, showing that stereo audio supports beyond-sight search while vision enables accurate final manipulation.

**Audio representation.** Table I-(b) compares different audio representations. Log-mel spectrograms show poor performance with high variance across random seeds, suggesting that they are less effective at preserving the spatial cues needed for reliable sound localization. The 4-channel spectrogram, which includes both magnitude and phase for the left and right channels, performs even worse. We attribute this to the sensitivity of phase information to reflections, reverberation, and geometric variation from robot motion, which makes it unstable in real-world settings. In contrast, our 2-channel magnitude spectrogram achieves the best performance.

### D. Generalization to Unseen Environments

Fig. 3-(b), (c), (d) and Table I-(c), (d), (e) present the qualitative and quantitative results, respectively. We construct

TABLE I: Evaluation on the *beyond-sight ring-off* task. Success rates are reported as mean and standard deviation over three random seeds. Precise Success Rate (PSR) counts only cases where the robot accurately presses the red tag to turn off the target.

| Method | SR (↑) | PSR (↑) |
|---|---|---|
| **Evaluation 1: In-Distribution** | | |
| *(a) Modalities* | | |
| Vision-Only | $0.27 \pm 0.21$ | $0.13 \pm 0.15$ |
| Audio-Only (Stereo) | $0.73 \pm 0.15$ | $0.17 \pm 0.12$ |
| Audio-Visual (Mono) | $0.67 \pm 0.15$ | $0.47 \pm 0.15$ |
| Stereo Audio-Visual (Ours) | $\mathbf{1.00 \pm 0.00}$ | $\mathbf{0.70 \pm 0.20}$ |
| *(b) Audio Representation* | | |
| Log-Mel Spectrogram | $0.37 \pm 0.31$ | $0.27 \pm 0.12$ |
| 4-Ch Spectrogram (Mag & Phase) | $0.17 \pm 0.06$ | $0.13 \pm 0.06$ |
| 2-Ch Spectrogram (Ours) | $\mathbf{1.00 \pm 0.00}$ | $\mathbf{0.70 \pm 0.20}$ |
| **Evaluation 2: Unseen Audio Distractors** | | |
| *(c) Audio Augmentation* | | |
| No Audio Augmentation | $0.20 \pm 0.10$ | $0.13 \pm 0.06$ |
| Background Audio Augmentation | $0.77 \pm 0.21$ | $0.33 \pm 0.15$ |
| Spatial Audio Augmentation (Ours) | $\mathbf{0.80 \pm 0.10}$ | $\mathbf{0.40 \pm 0.10}$ |
| **Evaluation 3: Unseen Visual Distractors** | | |
| *(d) Cluttered Scene with a Silent Decoy* | | |
| Audio-Only (Stereo) | $\mathbf{0.60 \pm 0.20}$ | $0.07 \pm 0.12$ |
| Stereo Audio-Visual (Ours) | $0.40 \pm 0.30$ | $\mathbf{0.23 \pm 0.15}$ |
| **Evaluation 4: Unseen Initial Poses** | | |
| *(e) Random Initial Poses* | | |
| Audio-Only (Stereo) | $0.53 \pm 0.12$ | $0.07 \pm 0.06$ |
| Stereo Audio-Visual (Ours) | $\mathbf{0.80 \pm 0.10}$ | $\mathbf{0.43 \pm 0.06}$ |

three unseen environments to evaluate generalization.

**Unseen audio distractors.** Fig. 3-(b) and Table I-(c) evaluate robustness to unseen audio distractors. We consider five types of distractors: classical music as an irregular acoustic pattern, conversation and keyboard typing as everyday noise, alarm clock as a high-frequency distractor, and a similar beep as the most challenging case due to its temporal similarity to the target. For each distractor type, we perform two trials, one from the left and one from the right, resulting in 10 trials in total. Without audio augmentation, the policy performs poorly, suggesting that it overfits to simple loudness-based cues. Background audio augmentation substantially improves performance by exposing the policy to competing sounds during training. Spatial audio augmentation further provides modest but consistent gains in both SR and PSR, indicating that varying the inter-channel balance of distractors helps the policy better distinguish the target sound from unseen competing sounds.

**Unseen visual distractors.** Fig. 3-(c) and Table I-(d) evaluate robustness in a highly cluttered scene. We introduce many unseen objects as visual distractors and place a silent decoy object with the same appearance along the path to the sounding target. This setting is particularly challenging: over-reliance on vision may cause the policy to stop at the silent decoy, whereas over-reliance on audio may lead to imprecise turn-off. Although the overall success rate decreases under severe visual clutter, our policy achieves higher precise

success than Audio-Only, indicating that vision remains important for accurate final interaction once the target becomes visible. However, many failures are caused by timeouts. In these cases, the visually same decoy causes temporary hesitation, and the episode ends before successful turn-off even when the robot eventually moves toward the correct target. Audio-Only achieves a higher SR because it is not distracted by visual clutter, but its PSR remains much lower, highlighting the role of vision in precise final interaction.

**Unseen initial poses.** Fig. 3-(d) and Table I-(e) evaluate robustness to unseen initial poses. To test generalization to out-of-distribution robot starting states, we randomly vary the initial end-effector position over a wider region spanning 15 cm along the x-axis and 80 cm along the y-axis. This produces several challenging cases not seen during training, including farther starting distances, already-in-sight cases, and vertically aligned approaches in which the robot should move almost directly forward. Overall, the policy demonstrates robustness under these unseen initial conditions. As illustrated in Fig. 3-(d), even when the object is vertically aligned with the agent, the policy successfully identifies the direction of the target by employing lateral exploratory scanning. This allows it to break the symmetry and reorient toward the sound source, suggesting that the policy has learned a generalized relative seeking strategy rather than overfitting to specific trajectories. Failure cases primarily stem from timeouts or proximity to safety boundaries, as the increased initial variance occasionally constrains the available time for corrective maneuvers.

## V. Conclusion

We propose `SoundAct`, a spatial sound-aware egocentric robot manipulation framework that leverages stereo microphones and a wrist-mounted camera to address beyond-sight manipulation challenges. Our results show that stereo audio provides reliable spatial guidance before the target becomes visible, improving robustness under partial observability, distractors, and varying initial conditions, including unseen audio distractors, visual distractors, and random initial poses. Since real-world environments are inherently structured around human sensory capabilities, we believe this work highlights the value of stereo audio in robot manipulation as a cue for spatial awareness and represents a meaningful step toward more robust robots for human-centric everyday settings.

**Limitations and future works.** While `SoundAct` shows strong performance in controlled settings, its performance degrades in the presence of audio and visual distractors, indicating room for improved robustness. Future work will evaluate the framework in more realistic in-the-wild environments, incorporate short-term audio memory to handle transient sound sources, and extend the task to broader settings with more diverse action spaces.

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
