# OpenReview forum: "SoundAct: Learning Spatial Sound Awareness for Egocentric Robot Manipulation with Stereo Audio"
_IEEE.org/ICRA/2026/Workshop/Manipulation_Robustness — ICRA 2026_

### Official Review · Reviewer_tE9K · 2026-05-17
**Well-motivated stereo-audio manipulation framework with a clean modality ablation; the empirical scope and task design limit the strength of the conclusions.**

**Rating:** 7
**Confidence:** 4

**Review:**

Approach: This paper proposes SoundAct, a multimodal manipulation policy that integrates stereo audio with an wrist camera to enable beyond-sight manipulation. Stereo audio is encoded as 2-channel magnitude spectrograms and fused with CLIP-ViT visual features through a self-attention transformer, with a Diffusion Policy action head for action prediction. A spatial audio augmentation injects distractor noise with complementary per-channel scaling to prevent overfitting to absolute loudness. The framework is evaluated on a beyond-sight ring-off task across four axes: in-distribution performance, unseen audio distractors, unseen visual distractors, and unseen initial poses.

Strengths:
- The paper is well written, with an informative and progressively framed introduction of an underexplored audio modality for guiding policies on out-of-sight tasks. The problem is well motivated and the experimental design is detailed and considerate in verifying the method's robustness.
- In Section IV-D, the authors present a detailed and controlled ablation that varies visual and audio distractors as well as initial position, which makes the evaluation systematic and compelling in demonstrating the method's robustness under multiple common and meaningful variations.
- As shown in Table I, the proposed method achieves on average considerable improvements in both SR and PSR across most variations and experimental settings, making the method empirically compelling.

Improvements:
- The term "egocentric camera" used throughout the paper is imprecise and risks confusion with egocentric human data (e.g., Egoverse, Egoscale) where the camera is head-mounted on a human observer. Referring to it consistently as a wrist-mounted camera would improve clarity and avoid conflation with the broader egocentric vision literature.
- The definition of SR and PSR is imprecise. In Section IV-B, SR is defined as successfully turning off the target alarm clock within the time limit, while PSR counts only cases in which the robot accurately presses the red tag within the same time limit. However, the red tag appears to cover only part of the alarm button, and pressing outside the red region should still stop the alarm. If the authors want to evaluate manipulation precision rigorously, the task could be revised with an object that genuinely requires precise contact, such that spatial audio alone would almost always fail. The current design does not strongly support the Section IV-C conclusion that stereo audio alone is insufficient for precise turn-off. A complementary ablation that increases the number of spatial microphones, rather than adding vision, would also help isolate whether the precision gap is due to the modality itself or to the information richness of the stereo configuration.
- The unseen-environment studies in Section IV-D are limited in that the overall task environment remains unchanged (same table, same room, presumably similar lighting). This part would benefit from running the same task in a meaningfully different environment. Additional results with different stereo-microphone configurations would further strengthen the claim that the method is robust to microphone choice rather than tuned to the specific hardware used.

---

### Decision · Program_Chairs · 2026-05-21

Accept